# Inductive Relation Inference of Knowledge Graph Enhanced by Ontology Information

**Wentao Zhou**[1,*] **Jun Zhao**[1,*] **Tao Gui**[2,†] **Qi Zhang**[1,†] **Xuanjing Huang**[1,3]

[1]School of Computer Science, Fudan University
[2]Institute of Modern Languages and Linguistics, Fudan University
[3]International Human Phenome Institutes (Shanghai)
{zhaoj19,tgui,qz}@fudan.edu.cn, wtzhou22@m.fudan.edu.cn

## Abstract

The inductive inference of the knowledge graph aims to complete the potential relations between the new unknown entities in the graph. Most existing methods are based on entity-independent features such as graph structure information and relationship information to inference. However, the neighborhood of these new entities is often too sparse to obtain enough information to build these features effectively. In this work, we propose a knowledge graph inductive inference method that fuses ontology information. Based on the enclosing subgraph, we bring in feature embeddings of concepts corresponding to entities to learn the semantic information implicit in the ontology. Considering that the ontology information of entities may be missing, we build a type constraint regular loss to explicitly model the semantic connections between entities and concepts, and thus capture the missing concepts of entities. Experimental results show that our approach significantly outperforms large language models like ChatGPT on two benchmark datasets, YAGO21K-610 and DB45K-165, and improves the MRR metrics by 15.4% and 44.1%, respectively, when compared with the state-of-the-art methods.

## 1 Introduction

Knowledge graphs (KGs) store a large amount of structured real-world knowledge through a set of triples, and they have been widely used in many domains, such as natural language processing (Zhang et al., 2019), recommendation systems (Wang et al., 2018), and question answering (Huang et al., 2019). However, even the most knowledge-rich KGs suffer from incompleteness, such as DBpedia (Lehmann et al., 2015), YAGO (Mahdisoltani et al., 2014), and WikiData (Vrandečić, 2012). To complete the KGs, knowledge graph inference aims to predict the missing links between entities in KGs.

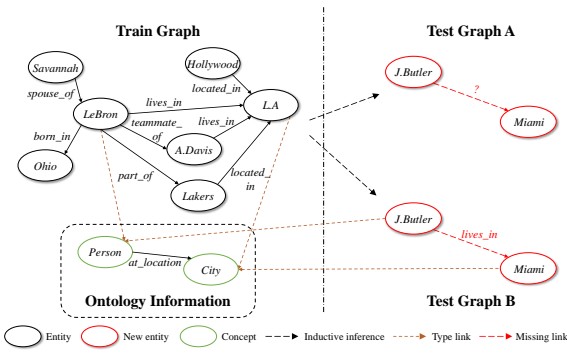

Figure 1: Two examples of inductive inference. As shown in Test Graph A, structure-based inductive inference methods have difficulty predicting relations between newly added entities that lack sufficient neighborhood information. Our work helps achieve inductive relation inference by using ontology information (Test Graph B).

In past studies, most methods (Bordes et al., 2013; Yang et al., 2015) mainly learn the specific embeddings of entities and relations and predict missing links by various mapping operations. Since the embeddings in such methods depends on specific entities, it requires that the entities in the graph are fixed, which is referred to as the transductive setting (Yang et al., 2016). However, in fact, new emerging entities are continuously added to real-world KGs over time, such as new users and products in e-commerce knowledge graphs and new molecules in biomedical knowledge graphs (Trivedi et al., 2017). Works adopting the transductive setting often require expensive retraining to make predictions for these added entities. As the amount of data increases this overhead will become unaffordable (Schlichtkrull et al., 2018). To deal with this problem, inductive inference gains the ability to extend what is learned from training entities to unknown entities by learning entity-independent semantic information (Teru et al., 2020).

However, most of the existing inductive infer-

---

*Equal Contributions.
†Corresponding authors.

ence work (Chen et al., 2021; Mai et al., 2021) has only focused on various structural features in KGs (e.g., enclosing subgraphs induced by paths between nodes), ignoring the important ontology information. In general, newly emerging entities tend to lack sufficient neighbor relationships, leaving them without much contextual information to refer to in terms of structure (As shown in Test graph A in fig. 1) (Xu et al., 2022). As an abstract description of the real world, various concepts in ontology provide the basic type information for the affiliated entities, which can help the model achieve inductive link prediction. For example, in Test Graph B, despite lacking the neighbor relationship, with the help of the type information provided by the concepts "$Person$" and "$City$" for "$J.Butler$" and "$Miami$" respectively, we can predict that the relation between them is most likely to be "$lives\_in$". In fact, the ontology information of entities also suffers from missing problems, so it is not easy to use ontology information effectively.

To solve the above problem, we propose a knowledge graph inductive inference method combining ontology information. Specifically, based on the enclosing subgraph, we bring in feature embeddings of corresponding concepts at the node initialization of entities to obtain the semantic information in the ontology. To deal with the problem of missing ontology information, we build a type constraint regular loss that captures the missing concepts of entities by explicitly modeling the semantic associations between entities and concepts. In addition, we train the link prediction on the ontology triples. The final training is performed using a joint strategy. Experimental results show that the method significantly outperforms ChatGPT as well as state-of-the-art inductive baselines. Our codes are publicly available at GitHub.*

Our contributions can be summarized in the following three points: (1) we propose a knowledge graph inductive inference method combining ontology information, which effectively improves the inductive inference performance on newly emerging entities; (2) we build type-constrained regular loss to alleviate the problem of missing ontology information; (3) we achieve a remarkable improvement on two benchmark datasets, demonstrating the effectiveness of using ontology information to enhance the effectiveness

---

*https://github.com/chasers-of-Qs/OEILP

of inductive link prediction.

## 2 Related Works

**Transductive Link Prediction**: Most existing knowledge graph inference methods are embedding-based transductive learning. These methods can be broadly classified into: 1) translation-based(Bordes et al., 2013; Wang et al., 2014; Lin et al., 2015; Sun et al., 2019); 2) semantic matching-based, (Yang et al., 2015; Trouillon et al., 2016; Nickel et al., 2016); 3) GNN-based(Schlichtkrull et al., 2018; Vashishth et al., 2020; Liu et al., 2021). The difference between them mainly lies in how to design the score function.

**Inductive Link Prediction**: As inductive inference models have the ability to extend from known entities to unknown entities, these methods(Ali et al., 2021; Gesese et al., 2023) show great potential for link prediction tasks on new entities. BLP(Daza et al., 2021) learns the embedding representations of entities based on the architecture of the pre-trained language model to obtain the required generalization capabilities. GraIL(Teru et al., 2020) suggests modeling enclosing subgraphs around the target triple for the first time, based on the graph neural network framework. TACT(Chen et al., 2021) models the semantic correlation between relations as several topological patterns and uses a relational correlation network (RCN) to learn the importance of different patterns for inductive link prediction. SNRI(Xu et al., 2022) enhances the processing for sparse subgraphs by exploiting full neighbor relationships and by applying mutual information (MI) maximization to knowledge graphs.

**Ontology Enhanced Inference**: Incorporating ontology information through various methods(Xie et al., 2016; Ren et al., 2021), it helps models learn richer semantic information. TransC (Lv et al., 2018) models the embedding of concepts as a sphere and assumes that the embedding corresponding to the entity belonging to the concept should be in this sphere. Considering the limitation of relation information in the ontology, TransO (Li et al., 2023) computes the weights of the type mapping matrix based on domain and range. JOIE (Hao et al., 2019) proposes the first approach of jointly embedding entity and ontology knowledge graphs to build a unified representation learning framework from multiple levels. These

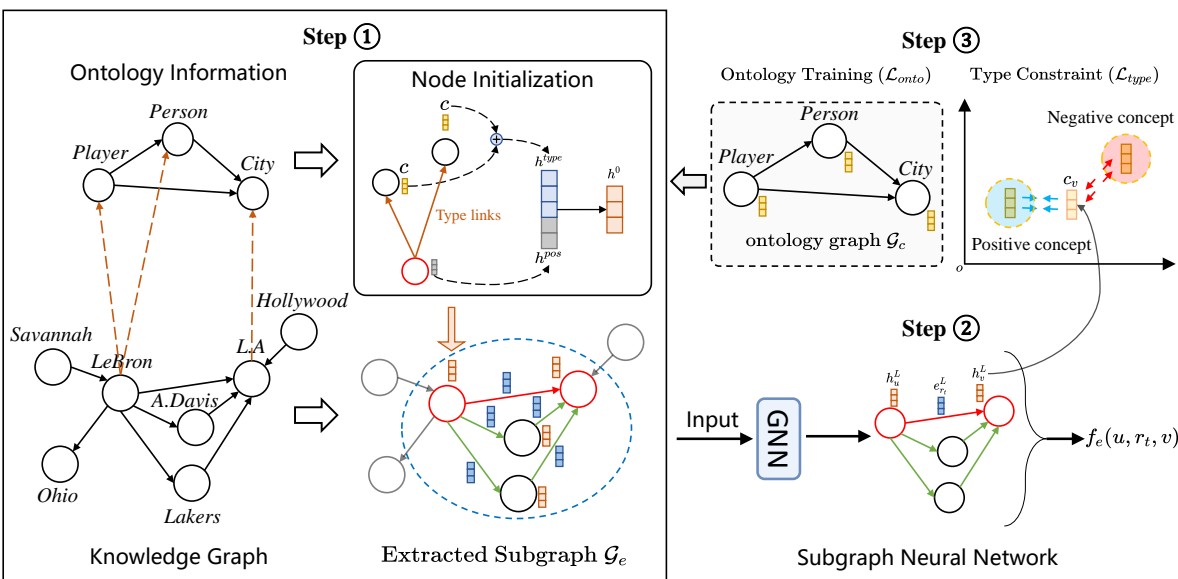

Figure 2: Overview of the proposed method. The training pipeline consists of two parts: extracting the enclosing subgraph of the target triple from KG and initializing the node embeddings with ontology information (**Step①**); optimizing the score function $f_e$ used to infer relations between unknown entities (**Step②**) In addition, we optimize the embedding of the ontology by ontology training and construct a type constraint to capture the semantic connections between concepts and entities (**Step ③**).

methods are all transductive link predictions on the original KGs. Inductive link prediction for unknown entities is still an urgent but under-researched task.

## 3 Approach

We start with task formulation. Unlike transductive link prediction which predicts unknown triples on a fixed entity graph, for inductive link prediction, the goal is to predict unknown triples $(u, r_t, v)$ in the unseen entity graph by learning from the seen entity graph. The entities in these two graphs are disjoint. Specifically, given the seen entity graph $\mathcal{G}_{e_k}$ and the migratable information (e.g., ontology graph $\mathcal{G}_c$ and type links between entities and concepts $\mathcal{T}_t$), we optimize the score function $f(s, r, t)$ at training time, where the score reflects the likelihood of the existence of target relation $r$ between target nodes $s$ and $t$. In testing, combined with the migratable information, we use the score function $f$ to predict the unknown triples $(u, r_t, v)$ in the unseen entity graph $\mathcal{G}_{e_u}$.

### 3.1 Method Overview

In this work, we propose an inductive link prediction method that combines entity and ontology information. The method brings in ontology information to enhance inductive inference on

unknown entities and effectively captures the missing ontology information of entities by type constraint. As shown in fig. 2, the whole training consists of three steps:

**Step ①:** This part aims to integrate ontology information. First, we extract the enclosing subgraph around the target node and obtain the position feature $h_i^{pos}$ of nodes in the subgraph. Second, we obtain the type feature $h_i^{type}$ from the embedding of the concept corresponding to the entity by the attention approach. Finally, we concatenate the position feature $h_i^{pos}$ with the type feature $h_i^{type}$ as the initial feature of the node. In this way, we integrate the semantic information of the ontology into the entity.

**Step ②:** This step aims to optimize the score function $f_e(u, r_t, v)$ used for prediction. For the representations of nodes and relations in the subgraph, we use a graph neural network (GNN) to update them. Then, we optimize the score function $f_e(u, r_t, v)$ obtained with these representations. Since the node representations contain type features, the model can use ontology information to help complete inductive link prediction.

**Step ③:** This step aims to construct type constraint and optimize ontology embedding. We construct a type constraint regular loss $f_t$ by modeling the relationship between entity embeddings and

concept embeddings. The constraint requires that the embedding of the entity $h$ should be close to the embedding of its corresponding concept $c$ after mapping to the feature space of the ontology. Based on this constraint, the model gains the ability to capture the missing ontology information of the entity. In addition, we train the link prediction task on the ontology triples to optimize the embedding representation of the ontology in order to enhance the semantic information obtained by the model from the ontology.

Finally, based on the above steps, the model learns from the total loss $\mathcal{L}$. Next, we will describe the technical details of each step in detail.

### 3.2 Ontology Information Feature Embedding

**Subgraph Extraction:** We extract the enclosing subgraph around the target triple $(u, r_t, v)$ from GraIL (Teru et al., 2020). First, we obtain the sets $N_k(u)$ and $N_k(v)$ of nodes in the corresponding $k$-hop neighborhoods from the two target nodes $u$ and $v$, respectively. Then, the duplicate nodes are removed by taking the intersection set $N_k(u) \cap N_k(v)$. Finally, the nodes isolated from any node or at a distance greater than $k$ are cut off to obtain the enclosing subgraph.

**Node Initialization:** We use the position feature $h_i^{pos}$ of the node and the type feature $h_i^{type}$ as the initial embedding of the node features to ensure that the node features do not contain any node attributes. First, we obtain the position feature $h_i^{pos}$ of the node in subgraph by double radius vertex labeling (Zhang and Chen, 2018) scheme:

$$h_i^{pos} = [\text{one-hot}(d(i, u)) \oplus \text{one-hot}(d(i, v))], \tag{1}$$

where $d(i, u)$ denotes the shortest distance from node $i$ to $u$ without passing through $v$. Second, we obtain the type feature $h_i^{type}$ from the embedding of the concept corresponding to the entity by the attention approach:

$$h_i^{\text{type}} = \sigma_1 \left( \sum_{c_j \in \mathcal{C}_i} \alpha_j W_1 c_j + b_1 \right) \tag{2}$$

$$\alpha_j = \text{softmax}(c_j, c) = \frac{\exp(c_j^{\mathrm{T}} c)}{\sum_{c_k \in \mathcal{C}_i} \exp(c_k^{\mathrm{T}} c)}, \tag{3}$$

where $\mathcal{C}_i$ is the set of concepts corresponding to node $h_i$, $c$ denotes the type relation between entities and concepts, $\alpha_j$ reflects the importance of the type information in each concept $c_j$ under the type

relation $c$, and $\sigma_1$ is the sigmoid function. Finally, by connecting the position feature $h_i^{pos}$ and type feature $h_i^{type}$, we obtain the initial embedding of the node $h_i^0$:

$$h_i^0 = [h_i^{type} \oplus h_i^{pos}]. \tag{4}$$

We think that using ontology type information to guide the node initialization helps the model to learn the semantic information implicit in the ontology.

### 3.3 Subgraph Neural Network

We input the enclosing subgraph $\mathcal{G}_{(u,r_t,v)}$ into the GNN to update the embedding of the nodes. We define the update function based on the architecture of R-GCN (Schlichtkrull et al., 2018):

$$h_t^k = \text{ReLU}(W_{self}^k h_t^{k-1} + a_t^k), \tag{5}$$

where $a_t^k$ denotes the neighbor feature aggregation function. Inspired by CompGCN (Vashishth et al., 2020) and edge attention, we define $a_t^k$ as:

$$a_t^k = \sum_{r=1}^{R} \sum_{s \in \mathcal{N}_r(t)} \alpha_{rr_tst}^k W_r^k \phi(e_r^{k-1}, h_s^{k-1}) \tag{6}$$

$$\alpha_{rr_tst}^k = \sigma_2(W_2^k s + b_2^k) \tag{7}$$

$$s = \text{ReLU}(W_3^k[h_s^{k-1} \oplus h_t^{k-1} \oplus e_r^{k-1} \oplus e_{r_t}^{k-1}] + b_3^k), \tag{8}$$

where $\mathcal{N}_r(t)$ denotes the direct outgoing neighbors of node $t$ under relation $r$, $\alpha_{rr_tst}^k$ is the edge attention weight of edge $(s, r, t)$ at layer $k$, and $\sigma_2$ is the sigmoid function. $\phi(e_r^{k-1}, h_s^{k-1})$ is the fusion operation on the features of the neighboring nodes and relations. We set it as the subtraction $\phi(\boldsymbol{e}, \boldsymbol{h}) = \boldsymbol{h} - \boldsymbol{e}$ (Vashishth et al., 2020). In order to keep the same embedding space for nodes and relations, we also update the relation embedding:

$$e_r^k = W_{rel}^k e_r^{k-1}, \tag{9}$$

In addition, we also use the JK-connection mechanism (Xu et al., 2018) on the representation of nodes and relations, and this approach makes the performance of the model robust to the number of layers of the GNN. The representation of the subgraph $\mathcal{G}_{(u,r_t,v)}$ is obtained by average pooling of all node representations:

$$h_{\mathcal{G}_{(u,r_t,v)}}^L = \frac{1}{|\mathcal{V}|} \sum_{i \in \mathcal{V}} h_i^L, \tag{10}$$

Finally, we obtain the score of the target triple $(u, r_t, v)$:

$$f_e(u, r_t, v) = W^T[h_{\mathcal{G}_{(u,r_t,v)}}^L \oplus h_u^L \oplus h_v^L \oplus e_{r_t}^L],$$

$$(11)$$

where $h_u^L$, $h_v^L$, and $e_{r_t}^L$ denote the embedding of target nodes $u$, $v$ and target relation $r$. We obtain the negative triples used in loss functions below by replacing the head or tail entity with uniformly sampled random entities. The margin-based loss function in the subgraph is:

$$\mathcal{L}_{ent} = \sum_{(u,r_t,v)\in\mathcal{G}_e} \max(0, f_e(u', r_t, v')$$
$$- f_e(u, r_t, v) + \gamma_1). \quad (12)$$

where $(u, r_t, v)$ and $(u', r_t, v')$ denote positive and negative samples, respectively, and $\gamma_1$ is the margin hyperparameter.

### 3.4 Type Constraint and Ontology Training

We explicitly model the relationship between entity embeddings and concept embeddings. Specifically, this requires that the embedding of the entity $h$ should be close to the embedding of its corresponding concept $c$ after mapping to the ontology embedding space. This way builds a type constraint $f_t$ in the form of regularization:

$$f_t(u, v, \mathcal{C}_{u,v}) = \frac{1}{|\mathcal{C}_{u,v}|} \sum_{c_w\in\mathcal{C}_{u,v}} \|c_w - c_{u,v}\|_2 \quad (13)$$

$$c_{u,v} = \sigma_3(W_4 h_{u,v} + b_4), \quad (14)$$

where $h_{u,v}$ is the embedding of the target node $u$ or $v$, $c_{u,v}$ is the embedding of the entity embedding after mapping to the ontology embedding space, $c_w$ is the embedding of the corresponding concept of node $u$ or $v$, and $\sigma_3$ is the tanh function. We use the same negative sampling method as before. Thus, the margin-based type constraint regular loss is:

$$\mathcal{L}_{type} = \sum_{(u,r_t,v)\in\mathcal{G}_e} \max(0, f_t(u, v, \mathcal{C}_{u,v})$$
$$- f_t(u, v, \mathcal{C}_{u,v}') + \gamma_2). \quad (15)$$

where $(u, v, \mathcal{C}_{u,v})$ and $(u, v, \mathcal{C}_{u,v}')$ denote positive and negative samples. Inspired by(Hao et al., 2019; Dong et al., 2021), learning meta-relations between concepts[†] will enhance the semantic

---

[†]Meta-relations include hierarchical relations between concepts and other general meta-relations

| Dataset | | #E | #R | #ET | #C | #M | #OT | #TL |
|---|---|---|---|---|---|---|---|---|
| **YAGO21K-610** | train | 16357 | 30 | 30000 | 610 | 24 | 1983 | 4861 |
| | valid | 4388 | 21 | 3000 | 166 | 14 | 248 | 1783 |
| | test | 3938 | 25 | 6970 | 159 | 13 | 248 | 1898 |
| **DB45K-165** | train | 29569 | 230 | 66000 | 165 | 20 | 516 | 29569 |
| | valid | 10165 | 189 | 6600 | 53 | 8 | 65 | 10166 |
| | test | 9681 | 177 | 15000 | 51 | 8 | 65 | 9682 |

Table 1: Statistics of inductive benchmark datasets. We denote the number of entities, relations, entity triples, concepts, meta-relations, ontology triples, and type links using #R, #E, #ET, #C, #M, #OT, and #TL respectively.

information obtained by the model from the ontology. Therefore, we train the link prediction task on the ontology triplet $(u_c, m_t, v_c)$, and the corresponding score function is:

$$f_o(u_c, m_t, v_c) = \|h_{u_c} + m_t - h_{v_c}\|_2, \quad (16)$$

where $h_{u_c}$ and $h_{v_c}$ denote the embedding of concepts, and $m_t$ denotes the embedding of meta-relations between concepts. The margin-based loss function obtained by ontology training is:

$$\mathcal{L}_{onto} = \sum_{(u_c,m_t,v_c)\in\mathcal{G}_c} \max(0, f_o(u_c, m_t, v_c)$$
$$- f_o(u_c', m_t, v_c') + \gamma_3). \quad (17)$$

where $(u_c, m_t, v_c)$ and $(u_c', m_t, v_c')$ denote positive and negative samples.

### 3.5 Joint Training Strategy

We combine all the losses to obtain the total loss $\mathcal{L}$. The overall training objective is as follows:

$$\mathcal{L} = \mathcal{L}_{ent} + \alpha\mathcal{L}_{ont} + \omega\mathcal{L}_{type}. \quad (18)$$

where $\alpha$ and $\omega$ are weighting hyperparameters. With the joint training strategy, our model can better utilize the ontology information to enhance the inductive inference on unknown entities, while effectively capturing the missing ontology information of entities.

## 4 Experimental Setup

### 4.1 Datasets

The KG benchmark datasets with ontology information, YAGO26K-906 and DB111K-174 (Hao et al., 2019), are originally developed for transductive inference prediction. To facilitate the inductive test, we build two new datasets, **YAGO21K-610** and **DB45K-165**, based on the two original benchmark datasets. Both datasets are built from entity triples,

| Method | YAGO21K-610 | | | DB45K-165 | | |
|---|---|---|---|---|---|---|
| | MRR↑ | Hits@1↑ | Hits@10↑ | MRR↑ | Hits@1↑ | Hits@10↑ |
| GraIL(Teru et al., 2020) | 0.682 | 0.666 | 0.684 | 0.540 | 0.515 | 0.548 |
| TACT(Chen et al., 2021) | 0.688 | 0.674 | 0.689 | 0.493 | 0.471 | 0.485 |
| SNRI(Xu et al., 2022) | 0.384 | 0.332 | 0.528 | 0.408 | 0.316 | 0.556 |
| ChatGPT | 0.271 | 0.219 | 0.333 | 0.504 | 0.431 | 0.603 |
| **Ours** | **0.794(15.4)** | **0.758(12.5)** | **0.845(22.6)** | **0.778(44.1)** | **0.704(36.7)** | **0.925(53.4)** |

Table 2: Main results on two inductive benchmark datasets. The numbers in **bold** and underlined indicate the best and second-best results, respectively, and the numbers in ( ) indicate the percentage improvement of our method over the best baseline result.

ontology triples, and type links. In the entity triples, the entities in the test set do not appear in the train set and valid set, while the relations in both the test set and valid set are included in the train set. We train on the train graph and test on the test graph. In addition, to achieve ontology training, we randomly divide the ontology triples into a train set, a valid set, and a test set in the ratio of 80%/10%/10%. tab. 1 provides the complete statistics for both datasets.

Please refer to Appendix A.1 for the detailed generation process of the datasets.

## 4.2 Compared Methods

To evaluate the effectiveness of our proposed approach, we compare our method with several state-of-the-art baseline methods. In addition, large language models (e.g., ChatGPT, GPT-4, etc.) have recently achieved impressive performance on several natural language processing tasks. Therefore, we also compare it with ChatGPT.

**GraIL**(Teru et al., 2020). The earliest method for inductive inference based on graph neural networks. It uses locally enclosing subgraphs and entity-independent node labels to represent node embeddings.

**TACT**(Chen et al., 2021). An inductive inference method based on relational correlation networks. The method categorizes all relation pairs into several topological patterns and uses the topology of the knowledge graph to learn the semantic correlation between relations.

**SNRI**(Xu et al., 2022). An inductive inference method based on graph neural networks. It utilizes complete neighborhood relations in terms of both neighborhood relation features of node features and neighborhood relation paths of sparse subgraphs and also models the neighborhood relations using mutual information maximization.

## 4.3 Metrics

As in the previous work (Teru et al., 2020; Xu et al., 2022), for the test triples $(h, r, t)$, we combine head (or tail) entities and relations with 50 candidate tail (or head) entities (including the original tail (or head) entities) obtained by random sampling to get positive and negative triples and rank all triples based on their scores. We use three metrics widely used in link prediction for evaluation. (1) **MRR**: The average of the inverse of the ranking of the correct entities in all tested samples. (2) **Hits@1**: The ratio of correct entities ranked first in all test samples. (3) **Hits@10**: The ratio of correct entities ranked within the top ten for all test samples.

## 4.4 Implementation Details

We extracted 3-hop enclosing subgraphs. In the training process, we use Adam as the optimizer, the learning rate is set to 0.01, and the batch size is set to 16. We use a three-layer GNN, and the dimensions of all feature embeddings are 32 except for the dimensions of the type feature embedding which is 24. The margins in the loss function are set to 10,10,5, and the weighting hyperparameters $\alpha$ and $\omega$ are set to 1. The maximum number of training epochs is 30. All experiments are conducted with Python 3.8.12 and PyTorch 1.11.0, using a GeForce GTX 2080Ti with 12GB RAM.

## 5 Results and Analysis

### 5.1 Main Results

In this section, we make inductive link predictions for the proposed method and several comparative methods, and the results are shown in tab. 2. Since we evaluate on a newly constructed benchmark dataset, we re-implement GraIL, TACT, and SNRI under our evaluation metrics. To make a fair comparison, we keep the hyperparameters the

| Part | YAGO21K-610 | | | DB45K-165 | | |
|------|------|------|------|------|------|------|
| | MRR↑ | Hits@1↑ | Hits@10↑ | MRR↑ | Hits@1↑ | Hits@10↑ |
| Head | 0.853 | 0.816 | 0.934 | 0.774 | 0.705 | 0.915 |
| Tail | 0.734 | 0.700 | 0.755 | 0.781 | 0.703 | 0.935 |
| No type | 0.743 | 0.714 | 0.757 | 0.710 | 0.613 | 0.914 |
| Type | 0.846 | 0.803 | 0.936 | 0.787 | 0.716 | 0.926 |
| **Ours** | 0.794 | 0.758 | 0.845 | 0.778 | 0.704 | 0.925 |

Table 3: Comparison of prediction performance in different components. Head (or Tail) indicates the performance when predicting the tail entity (or head entity) given the head entity (or tail entity) and the relationship, respectively. Type and No type indicate the performance when the entity to be predicted has and does not have type information, respectively.

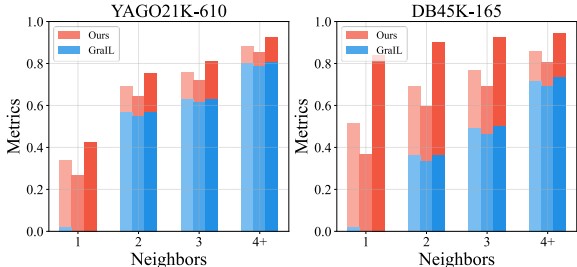

Figure 3: Prediction performance for target nodes with different number of neighboring nodes. When entities lack ontology information, our method degrades to equal the baseline method. In each set of results, MRR, Hits@1, and Hits@10 are shown from left to right.

same as their original papers. Similarly, we test ChatGPT with the same test data. The relevant details of the prompts are shown in Appendix A.2. From the results, we can observe that our method significantly outperforms the state-of-the-art baseline methods and ChatGPT. Specifically, some baseline methods (e.g., GraIL, TACT) only focus on various structural features in the knowledge graph, which makes them unable to handle sparse subgraphs effectively. And when the subgraph is empty (i.e., only the target triple exists), its performance even drops to the same as random guesses. In contrast, our approach effectively alleviates this problem by incorporating ontology information and achieves an overall performance improvement (We provide more results to support this claim in sec. 5.2). Although approaches like SNRI attempt to deal with the sparse subgraph problem by importing global information, our proposed method still outperforms them. Moreover, although ChatGPT outperforms some of the baseline methods (or even all of them) in some metrics, our method achieves better results compared to it.

In addition, we also make transductive link predictions for the proposed method and several baseline methods for further comparison. For detailed experimental results, please refer to Appendix A.3.

## 5.2 Result Analysis

In this section, we analyze the sources of performance improvement of our approach. First, we counted all the test triples, and the results are shown in tab. 3. We can observe that the prediction performance for entities with ontology type information outperforms that for entities lacking ontology information on both datasets.

Meanwhile, the prediction performance for tail entities outperforms that for head entities on the YAGO21K-610 dataset, while the prediction performance for both is similar on the DB45K-165 dataset. In addition, the proportion of head entities lacking type information is higher than that in tail entities on the YAGO21K-610 dataset, while the proportions are close on the DB45K-165 dataset. Therefore, we believe that the integration of ontology information improves the prediction performance of our model.

Furthermore, we counted the prediction performance for target nodes with different numbers of neighboring nodes. For the target nodes, the more neighboring nodes they have, the more paths may exist between the nodes and thus the more dense enclosing subgraphs are extracted. Therefore, by analyzing the performance improvement of the prediction of target nodes with different numbers of neighboring nodes by ontology information, we investigate the effect of ontology information on subgraphs with different densities, and the results are shown in fig. 3.From the results, we can see that integrating ontology information improves the prediction performance of target nodes with all different numbers of neighboring nodes, and the lower the number of neighboring nodes, the greater the performance improvement. This indicates that our method can improve the prediction for all enclosing subgraphs, and effectively alleviate the problem of poor prediction performance for sparse subgraphs at the same time.

## 5.3 Type Prediction

Our previous analysis of the experimental results demonstrates that ontology information can effectively improve prediction performance. In fact,

| Dataset | MRR↑ | Hits@1↑ | Hits@10↑ |
|---|---|---|---|
| **YAGO21K-610** | 0.858 | 0.791 | 0.953 |
| **DB45K-165** | 0.556 | 0.388 | 0.920 |

Table 4: Type prediction results on two inductive benchmark datasets.

ontology information also suffers from missing information, for example, on the test set of the YAGO21K-610 dataset, only 1725 entities have type information and more than half of them miss ontology information. Since the semantic links between entities and concepts are explicitly modeled in our method, we can use our method to predict type information for the test triples. Specifically, without providing ontology information, we predict the concept corresponding to the target node based on the enclosing subgraph of the target triple, and the prediction results are shown in tab. 4. From the results, we can see that our method can effectively predict the type information of entities, which helps to alleviate the problem of missing ontology information.

### 5.4  Ablation Study

To investigate the contribution of each component of our approach, we conduct ablation experiments on two datasets, and the experimental results are shown in tab. 5. First, like our analysis, the type information provided by the ontology plays a very important role in the inductive link prediction. When this module is removed, the prediction effect degrades to equal the baseline method, and the performance is severely compromised. Moreover, ontology training and type constraint improve the performance of the model in terms of optimizing the embedding representation of ontology and explicitly modeling the semantic connection between instances and concepts, respectively, and removing either of these modules leads to a degradation of performance. And when both modules are removed at the same time, the performance will be further degraded.

### 5.5  Hyper-parameter Analysis

As the core structure that is relied on when predicting, the size of the enclosing subgraph implies how much semantic information the target node can obtain from the structure. We conducted experiments on two datasets to investigate the effect of hop (which directly responds to the

| Method | MRR↑ | Hits@1↑ | Hits@10↑ |
|---|---|---|---|
| w/o type information | 0.664 | 0.649 | 0.665 |
| w/o ontology training | 0.788 | 0.729 | 0.842 |
| w/o type constraint | 0.770 | 0.722 | 0.837 |
| w/o both | 0.758 | 0.708 | 0.835 |
| **Ours** | **0.794** | **0.758** | **0.845** |
| w/o type information | 0.545 | 0.517 | 0.551 |
| w/o ontology training | 0.759 | 0.682 | 0.919 |
| w/o type constraint | 0.758 | 0.680 | 0.915 |
| w/o both | 0.717 | 0.648 | 0.868 |
| **Ours** | **0.778** | **0.704** | **0.925** |

Table 5: Abalation study of our method. The upper (resp. lower) part lists the results on YAGO21K-610 (resp. DB45K-165).

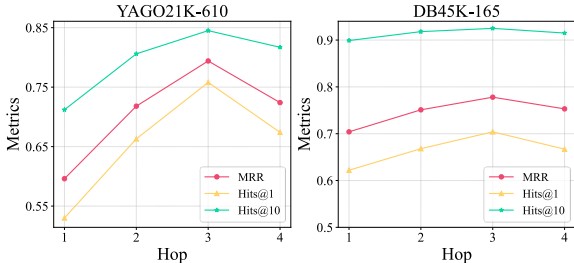

Figure 4: Inductive prediction performance of models with different hop.

size of the enclosing subgraph) on inductive prediction. From fig. 4, we can obtain the following observations. With the hop gradually increasing from 1, the prediction performance of the model keeps improving. This implies that too few neighbor nodes cannot provide enough semantic information for prediction(We provide more results in Appendix A.4 to support this claim). And when hop exceeds a certain threshold (i.e., 3), the performance starts to decrease. This indicates that the subgraph contains the critical structure needed for prediction after reaching a certain size and continuing to increase the size of the subgraph will continuously increase the number of noise nodes, which will lead to performance degradation.

### 5.6  Error Analysis

**What are the remaining errors in our research?**
For the inductive prediction of various subgraphs with ontology information, our method has brought different degrees of performance improvement. Moreover, our method can capture the missing ontology information of the target nodes through the rich semantic information in the subgraphs. However, for those enclosing subgraphs that lack ontology information and are extremely sparse,

our method has difficulty in giving correct type predictions due to the lack of available structural information, making the inductive prediction on such subgraphs not effectively improved. In fact, link prediction on extremely sparse enclosing subgraphs is difficult for all inductive prediction methods.

## 6 Conclusions

In this work, we propose a knowledge graph inductive inference method that combines ontology information. The approach integrates semantic information of ontology by using type information for the initialization of node features. We construct a type-constrained regular loss, which effectively captures the missing ontology information of entities. At the same time, ontology training helps the model to enhance the semantic information obtained from the ontology. Experimental results show that our approach achieves state-of-the-art inductive link prediction.

## Limitations

Although we demonstrate the effectiveness of ontology training for improving model performance, we only use the simplest methods to model ontology graphs. Using some richer and more effective methods (e.g., using pre-trained embedding representations or building detailed hierarchies) to learn better embedding representations of the ontology can help make the model further achieve superior performance. In addition, in all experiments, we set all margin and weighting hyperparameters as fixed hyperparameters, which may make our model not achieve its best performance. A dynamic optimization of these hyperparameters may be a better choice. We leave these limitations to our future work.

## Acknowledgements

The authors wish to thank the anonymous reviewers for their helpful comments. This work was partially funded by National Natural Science Foundation of China (No.62076069,62206057,61976056), Shanghai Rising-Star Program (23QA1400200), and Natural Science Foundation of Shanghai (23ZR1403500).

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

## A  Appendix

### A.1  Dataset Generation

To generate the train set, we randomly select a triple from the whole entity triples and treat it as the initial train graph, and keep adding new edges and nodes to the graph by the edges connected to the train graph. To ensure that the entities in the test set do not appear in the train set, we first remove the train set from the entire graph and then use the same approach to generate the test set, while requiring that the relations in the test set must appear in the train set. Finally, we generate the valid set using the same way as we generated the train set, and in addition, to ensure the inductive setting, all entities in the valid set do not appear in the test set. The triples in the train set, valid set, and test set do not intersect each other.

### A.2  Prompt details

Our prompt includes a one-shot example, task instruction, relevant information, questions and candidate entities. The prompt templates are shown in tab. 6.

Where one-shot example is used to standardize the format of the output. <Information> indicates the neighbor information and type information of the test triples, <Question> is the link prediction for the test triples, and <Possible answer> denotes the candidate entities.

**Information:**
B is C's father.
A is the grandfather of C.
**Question:**
'_' is B's father.
**Possible answers to '_' include:**
A; B; C

**Answer:**
A; C; B

Provide the top 10 answers in descending order based on the likelihood of correctness from possible answers. Please answer in the format of the example above:

**Information:**
<Information>
**Question:**
<Question>
**Possible answers to '_' include:**
<Possible answer>

**Information:**
B is the mother of A.
C is the grandmother of A.
**Question:**
C is the mother of '_'.
**Possible answers to '_' include:**
A; B; C

**Answer:**
B; A; C

Provide the top 10 answers in descending order based on the likelihood of correctness from possible answers. Please answer in the format of the example above:

**Information:**
<Information>
**Question:**
<Question>
**Possible answers to '_' include:**
<Possible answer>

Table 6: Prompt template for predicting the head entity (upper panel) and tail entity (lower panel) in triples.

### A.3  Transductive Prediction

FB15K-237 is the entity KG benchmark dataset widely used in many recent works, and GraIL has proposed four versions of variant datasets based on it. We used some of these variant datasets to compare our approach with several baseline methods. Since FB15K-237 contains only entity triples, we supplemented the type information for the dataset used based on WikiData, and the statistics of the supplemented dataset are shown in tab. 7. In addition, due to the lack of carefully built ontology triples, our method uses the version without the ontology training module (w/o OT) for transductive link prediction. The results are shown in tab. 8. As can be seen from the results, our method outperforms the baseline methods even for transductive link prediction.

| Dataset | | #Entities | #Relations | #Triples | #Concepts | #Type links |
|---------|-------|-----------|------------|----------|-----------|-------------|
| v1 | train | 1093 | 142 | 1993 | 218 | 1313 |
| | valid | 287 | 66 | 206 | 165 | 413 |
| | test | 301 | 68 | 205 | 153 | 434 |
| v2 | train | 1660 | 172 | 4145 | 304 | 2002 |
| | valid | 548 | 92 | 469 | 219 | 755 |
| | test | 562 | 107 | 478 | 236 | 737 |
| v3 | train | 2501 | 183 | 7406 | 396 | 3151 |
| | valid | 973 | 120 | 866 | 310 | 1282 |
| | test | 981 | 128 | 865 | 330 | 1312 |

Table 7: Statistics on the FB15k-237 dataset after adding type information.

| Method | v1 | | v2 | | v3 | |
|--------|-------|---------|-------|---------|-------|---------|
| | MRR↑ | Hits@1↑ | MRR↑ | Hits@1↑ | MRR↑ | Hits@1↑ |
| GraIL | 0.499 | 0.412 | 0.615 | 0.499 | 0.635 | **0.529** |
| SNRI | 0.490 | 0.381 | 0.584 | 0.461 | 0.618 | 0.494 |
| TACT | 0.475 | 0.378 | 0.565 | 0.451 | 0.585 | 0.457 |
| **Ours** | **0.506** | **0.415** | **0.623** | **0.512** | **0.636** | 0.526 |

Table 8: Results of transductive link prediction on the FB15k-237 dataset.

## A.4 The Impact of Neighboring Nodes

We studied the effect of different enclosing subgraph sizes on the prediction during the test, and the experimental results are shown in fig. 5. It can be seen that the smaller the enclosing subgraph extracted for the test triple, the worse its prediction performance is without changing any other conditions. The direct result of a smaller enclosing subgraph is that the target node has fewer neighboring nodes available. Therefore, we argue that too few neighbor nodes will impair the semantic information captured by the model from the structure and thus lead to a decrease in prediction performance.

## A.5 Ontology Prediction

As a kind of knowledge graph, the ontology graph also suffers from the problem of incompleteness. Like other jointly trained methods, our proposed method can conveniently complete the ontology

| Method | YAGO21K-610 | | | DB45K-165 | | |
|--------|-------------|---------|----------|-----------|---------|----------|
| | MRR↑ | Hits@1↑ | Hits@10↑ | MRR↑ | Hits@1↑ | Hits@10↑ |
| JOIE | 0.243 | 0.159 | 0.383 | 0.279 | 0.231 | 0.315 |
| **Ours** | **0.278** | **0.185** | **0.454** | **0.418** | **0.331** | **0.592** |

Table 9: Results of link prediction on the ontology graph.

graph while achieving the prediction of entities. We compared our method with the jointly trained baseline method (Hao et al., 2019) and the results are shown in tab. 9. It can be seen that even though we only use the simplest model to train ontology triples, our method still outperforms the state-of-the-art baseline method, where JOIE builds the hierarchy of ontology in detail and uses a more advanced training model.

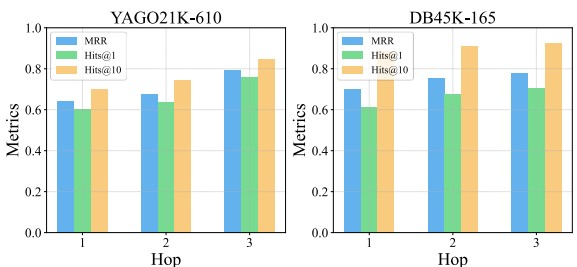

Figure 5: Prediction performance using enclosing subgraphs of different sizes.