# OpenReview forum: "Inductive Relation Inference of Knowledge Graph Enhanced by Ontology Information"
_EMNLP/2023/Conference — EMNLP 2023 Findings_

### Official Review · Reviewer_6Xgf · 2023-08-03

**Typos Grammar Style And Presentation Improvements:** Figure 1 could be made a bit larger.
**Soundness:** 4

**Excitement:**

4: Strong: This paper deepens the understanding of some phenomenon or lowers the barriers to an existing research direction.

**Missing References:**

https://arxiv.org/abs/2211.11407 \
https://arxiv.org/pdf/2010.03496.pdf \
https://arxiv.org/pdf/2107.04894.pdf

**Paper Topic And Main Contributions:**

The paper proposes a method for inductive inference on knowledge graphs. It not only considers unseen entities but also relations which is an under-explored area of study and very interesting. The proposed method considers ontological and type information for inductive link prediction and type prediction which is very interesting.

**Questions For The Authors:**

- In the ablation study, the difference between the proposed method with or without ontological information is not much different, the differences are only marginal. The reason should be explained. Does it show that ontological information does not help much?
- Does the inductive relational inference gets better when using ontological information?
- Why new datasets are introduced? Can the ontological and type information be induced in the existing datasets?

**Reasons To Accept:**

- The proposed approach based on ontology and the type of information is very interesting.
- The results are outperforming the algorithms reported in the paper but some of the related work is still missing.
- The paper is written very clearly.

**Reasons To Reject:**

- Missing related work which actually creates doubt if the approach is really effective in its experimental and theoretical comparison.

**Reproducibility:**

3: Could reproduce the results with some difficulty. The settings of parameters are underspecified or subjectively determined; the training/evaluation data are not widely available.

**Reviewer Confidence:**

4: Quite sure. I tried to check the important points carefully. It's unlikely, though conceivable, that I missed something that should affect my ratings.

---

> ### Author Rebuttal · Authors · 2023-08-27
>
> Thank you very much for your suggestions. We agree with the significance and value of the related work you mentioned in inductive link prediction. Since in the same dataset (such as FB15K-237), reference [1] significantly outperforms them on the same metrics, and our paper primarily considers comparing with several state-of-the-art baseline methods, we did not compare with these works.
>
> We appreciate you bringing up this idea. We will incorporate discussions about these works in our revised paper to address this concern.
>
> [1] Inductive Relation Prediction by Subgraph Reasoning
>
> Question regarding the role of ontology information (Q1&Q2):
>
> We would like to clarify that the use of ontology information significantly enhances the performance of inductive relation inference. In the results analysis of our paper (lines 458-496), we extensively examine the impact of ontology information on inductive relation reasoning. The experimental results show that the introduction of ontology information significantly enhances prediction performance, and this enhancement becomes more pronounced with fewer neighbor information. Moreover, ontology information is distinct from ontology training. Ontology information refers to the semantic information contained within the ontology, while ontology training is a module in our approach. As pointed out in our ablation study, the introduction of ontology information in our method includes three modules: type information, ontology training, and type constraint (lines 521-535). Specifically, the largest contribution comes from type information, and the absence of it severely impacts the performance. Although the performance drops from removing ontology training or type constraints alone is not as significant as removing type information, we note that removing either one or both leads to performance degradation (with more pronounced degradation when both are removed simultaneously). Thus, each module contributes to performance enhancement, varying only in the degree of contribution. We hope this addresses your concerns.
>
> Question regarding the introduction of a new dataset (Q3):
>
> Your question is very relevant. In response, we need to suggest that while type information can be induced from existing datasets using WikiData, ontology information still requires expensive manual construction by experts. The new dataset, being a KG benchmark dataset with ontology information, is more suitable in terms of data quality for our research. Although we have not explored the construction of ontology information for existing datasets, we acknowledge the potential value of such research. Therefore, we provide the existing dataset with type information in the supplementary material to facilitate the community in constructing its ontology information. We apologize for any confusion this might have caused.
>
> For any deficiencies or confusing language in the text or figures, we sincerely apologize. We will strive to address these issues to enhance the clarity and readability of the paper.
>
> Once again, we appreciate your valuable feedback, and we will carefully consider all your suggestions when revising the manuscript.

---

### Official Review · Reviewer_pao7 · 2023-08-04

**Soundness:** 3

**Excitement:**

3: Ambivalent: It has merits (e.g., it reports state-of-the-art results, the idea is nice), but there are key weaknesses (e.g., it describes incremental work), and it can significantly benefit from another round of revision. However, I won't object to accepting it if my co-reviewers champion it.

**Paper Topic And Main Contributions:**

The paper presents a KG link prediction approach that works for unknown entities (i.e., entities that were not there when a learning algorithm is trained). To enable ‘inductive learning’, authors propose to use both graph structure and ontology semantics in the learning process. For example, they infuse both type and positional information in learning node embeddings in the graph. Though it is simple, I like the idea of transforming entity embedding into ontology space and then enforce the similarity through an additional loss. The approach has 3 loss components; first is the normal entity space, margin-based loss and the other two are also margin-based but involves ontology space. However, use of ontology/schema information for KG embedding learning is not new (e.g., [1]). Using it for this specific problem with a pipeline of techniques can be also considered as an incremental improvement.

[1] Peng, Miao, Ben Liu, Qianqian Xie, Wenjie Xu, Hua Wang, and Min Peng. "SMiLE: Schema-augmented Multi-level Contrastive Learning for Knowledge Graph Link Prediction." arXiv preprint arXiv:2210.04870 (2022).


**Questions For The Authors:**

Question A: What is mt in Equations 16 and 17? I could not find its definition in the text. Also, I think this is similar to enforcing domain and range semantics of relationships. If so, how do you handle subclass of a class (that is the domain for example for the relationship) appearing as the type of an entity? Because then you have enumerate all the possibilities to not make learning algorithm penalize because exact string does not match for the triple you mention for example in Equation 16?

Question B: Could you elaborate on the point I mentioned earlier that use of schema/ontology for learning KG embeddings or link predictions is not new. Authors have mentioned some early works in this area. However, I am not sure by mentioning ‘inductive relation inference’ makes the paper’s contributions stand out. Could you express your thoughts?


**Reasons To Accept:**

•	Proposes a simple approach to incorporate ontology semantics for inductive link prediction.

•	Shows positive results over two benchmark datasets.



**Reasons To Reject:**

•	Using ontology information is not entirely exciting for KG link prediction tasks. Approach also does not propose any new technique other than the fusion of ontology semantics for the specific problem of inductive link prediction.

•	I think focus and contribution of the paper is on a narrow topic.


**Reproducibility:**

3: Could reproduce the results with some difficulty. The settings of parameters are underspecified or subjectively determined; the training/evaluation data are not widely available.

**Reviewer Confidence:**

4: Quite sure. I tried to check the important points carefully. It's unlikely, though conceivable, that I missed something that should affect my ratings.

---

> ### Author Rebuttal · Authors · 2023-08-27
>
> We appreciate your thorough review of our work and the insightful questions you raised. We would like to provide detailed responses to the questions you have raised as follows.
>
> Regarding the response to Question A:
>
> We agree with the question you pointed out. As mentioned in our approach (lines 329-331), (uc, mt, vc) represents the ontology triple, where uc and vc denote concepts, and mt denotes the meta-relation between concepts. In fact, this is different from enforcing domain and range semantics of relationships. In our paper, this part of the work aims to learn the semantic relationships (i.e., meta-relation) between concepts from ontology triplets to reinforce the semantic information within concepts. These meta-relations include hierarchical relationships between concepts (such as "is a" in (artist, is a, person) and "subclass of" in (movie, subclass of, art)) and other meta-relations (such as "at location" in (person, at location, city)). When an entity has multiple concepts and hierarchical relationships might exist between these concepts (e.g., "Einstein" has types "scientist" and "person"), another part of our work captures their different importance levels through the attention mechanism presented in Equations 2 and 3. We will revise the paper to clarify any confusion here and enhance the overall quality of the paper.
>
> Regarding the response to Question B:
>
> We understand your concerns regarding the contribution of our method and its distinction from other link prediction approaches that utilize ontology. In the introduction of our paper (lines 48-60), we mentioned that the real world represents an open-domain problem. Transductive link prediction requires costly retraining to handle the ever-growing new data, making this expense impractical as data volume increases. In contrast, inductive link prediction holds great value for knowledge graph completion due to its generalization capabilities. Furthermore, we note that the neighbor information of new entities in inductive inference is typically very sparse (lines 66-70). The introduction of ontology information helps address the issue of information scarcity in inductive relation inference, leading to significant improvements in predictive performance. While existing works have utilized ontology information for KG embedding learning, as we pointed out in related works (lines 155-159), previous works only focused on transductive link prediction. Our work stands out by significantly enhancing inductive relational inference by introducing ontology information. We will revise the paper to emphasize the focus and contributions of our proposed method. We hope this addresses your concerns.
>
> Thank you very much for taking the time to review our paper. For any parts of the text that may have caused confusion, we apologize and will work diligently to address these issues for improved clarity and readability of the paper.
>
> Once again, we appreciate your thoughtful comments.

---

### Official Review · Reviewer_TJtS · 2023-08-05

**Soundness:** 4

**Excitement:**

3: Ambivalent: It has merits (e.g., it reports state-of-the-art results, the idea is nice), but there are key weaknesses (e.g., it describes incremental work), and it can significantly benefit from another round of revision. However, I won't object to accepting it if my co-reviewers champion it.

**Paper Topic And Main Contributions:**

The paper addresses the crucial problem of inductive relation inference in knowledge graphs, specifically focusing on the sparsity of neighborhood information for new entities. The authors propose an innovative method that leverages ontology information to enhance the inductive inference of knowledge graphs.

**Questions For The Authors:**

1. This method achieves impressive results, but is the introduction of ontology information inconsistent with the nature of "unseen nodes" (especially in the test set)? It would be better to adding the result of the testing set without the type information.

2. The paper does not elaborate the details of ChatGPT in the paper. Does the prompt contain neighbor information and ontology information? Is it zero-shot or few-shot?


**Reasons To Accept:**

1. This paper proposes a novel and efficient approach that incorporates ontology information to improve the inductive inference of knowledge graphs.

2. The incorporation of the type-constrained regular loss is interesting, as it enhances the model's robustness.

3. The author improved the two datasets through a strict processing process, which is conducive to the experimental analysis of follow-up work.

4. The authors conduct a comprehensive experimental analysis.


**Reasons To Reject:**

1. As mentioned in Limitation Section, some modules of the approach can be replaced with more powerful alternatives, like initializing embedding of the node by BERT.

2. The experimental setting for comparing methods is unclear, and hyperparameter settings on different datasets should not be directly copied.


**Reproducibility:**

3: Could reproduce the results with some difficulty. The settings of parameters are underspecified or subjectively determined; the training/evaluation data are not widely available.

**Reviewer Confidence:**

4: Quite sure. I tried to check the important points carefully. It's unlikely, though conceivable, that I missed something that should affect my ratings.

---

> ### Author Rebuttal · Authors · 2023-08-27
>
> Thank you for your detailed comments and taking the time to review our work. We will address your concerns point by point.
>
> Question regarding the relationship between ontology information and the nature of "unseen nodes" (Q1):
>
> We would like to clarify that the introduction of ontology information is not contradictory to the nature of "unseen nodes." This is because in a knowledge graph, "unseen nodes" refer to entities that have not appeared during the training process. This "unseen" stems from the unknown of the entity itself and does not imply the absence of all information related to the entity. Even unknown entities can still possess ontology information. For example, in the triplet (J.Butler, lives in, Miami), even though we might not have encountered "J.Butler" before, we can still understand that he is a "Person." This ontology information aids us in inferring that the relationship between him and "Miami" is more likely to be "lives in" rather than other relationships like "is city of". In fact, as we mentioned in the ablation study (lines 521-526), the introduction of ontology information significantly improves the effectiveness of inductive link prediction compared to results without type information. We hope this helps address your question.
>
> Question regarding the details of ChatGPT (Q2):
>
> We agree with your perspective and would like to clarify the related details here. Our prompts include a one-shot example, task instruction, relevant information, question, and candidate entities. The one-shot example is used to standardize the format of the ChatGPT results. The task instruction is to rank the candidate entities based on the likelihood of being correct. The relevant information provides ontology and type information consistent with our method. The question pertains to the link prediction of the test triple, and the candidate entities are consistent with other methods. We deeply apologize for any confusion this may have caused and will modify the paper to supplement details related to ChatGPT.
>
> Thank you very much for taking the time to review our paper. We greatly appreciate your feedback and will carefully consider all of your suggestions when revising the manuscript.

---

### Meta-Review · Area_Chair_Dfaj · 2023-09-11

**Recommendation:** 2

**Metareview:**

This paper presents a methodology for a specific type of link prediction tasks, namely link prediction on graphs consisting of unseen entities (but with seen relations). The problem may have some relevance to data mining and knowledge management communities, whereas the significance for EMNLP is not clear (the only, weak link mentioned in the intro is that KGs are an ingredient in NLP tasks).

Reviewers found the paper generally sound, though excitement was limited. In particular, the problem was considered narrow, and several related works were not discussed. It appears the paper might be better suited for a data mining or knowledge management venue.

Further to the reviewer comments, I see three issues with this paper:
 - The reported problem appears strange - it is true that unseen entities are a problem in link prediction, but the formulation in this paper, of predicting on a completely disjoint graph, seems to misunderstand the problem: In the domain that the authors investigate (Wikidata-style knowledge), common general-world entities like countries, big cities, big companies definitely do re-occur between train and test sets. The challenge, instead, are organic entities that are only occuring a few times, but an (arbitrary?) hard split of the dense organic KG does not shed light on them.
 - The datasets are insufficiently explained, and seem extremely cherrypicked. There is no discussion of relations and instances to be predicted, and how the artificial splits were generated. MRR values around 0.8 represent outstanding ranking scores, and make me believe that the datasets consist of very strange instances, and are not indicative of realistic use cases. Instead of ranking metrics, it would be more helpful to report absolute or relative recall gains at a certain precision, e.g., how many, or what percentage of triples can the approach add to the KG, while maintaining 80% precision?
 - In addition to the strong absolute results (MRR), the relative gains (+15/+44% over SoA) are also at a level that is not easy to believe (extraordinary results require extraordinary evidence). The discussion provides zero case studies, and the metrics also aggregate across (unexplained) relations, thereby making it impossible to check them for plausibility.

---

### Decision · Program_Chairs · 2023-10-07

**Decision:**

Accept-Findings

**Comment:**

This paper presents a methodology for a specific type of link prediction tasks, namely link prediction on graphs consisting of unseen entities (but with seen relations). The problem may have some relevance to data mining and knowledge management communities, whereas the significance for EMNLP is not clear (the only, weak link mentioned in the intro is that KGs are an ingredient in NLP tasks).

Reviewers found the paper generally sound, though excitement was limited. In particular, the problem was considered narrow, and several related works were not discussed. It appears the paper might be better suited for a data mining or knowledge management venue.

Further to the reviewer comments, I see three issues with this paper:
 - The reported problem appears strange - it is true that unseen entities are a problem in link prediction, but the formulation in this paper, of predicting on a completely disjoint graph, seems to misunderstand the problem: In the domain that the authors investigate (Wikidata-style knowledge), common general-world entities like countries, big cities, big companies definitely do re-occur between train and test sets. The challenge, instead, are organic entities that are only occuring a few times, but an (arbitrary?) hard split of the dense organic KG does not shed light on them.
 - The datasets are insufficiently explained, and seem extremely cherrypicked. There is no discussion of relations and instances to be predicted, and how the artificial splits were generated. MRR values around 0.8 represent outstanding ranking scores, and make me believe that the datasets consist of very strange instances, and are not indicative of realistic use cases. Instead of ranking metrics, it would be more helpful to report absolute or relative recall gains at a certain precision, e.g., how many, or what percentage of triples can the approach add to the KG, while maintaining 80% precision?
 - In addition to the strong absolute results (MRR), the relative gains (+15/+44% over SoA) are also at a level that is not easy to believe (extraordinary results require extraordinary evidence). The discussion provides zero case studies, and the metrics also aggregate across (unexplained) relations, thereby making it impossible to check them for plausibility.